# Antifungal Action of Edible Coating Comprising Artichoke-Mediated Nanosilver and Chitosan Nanoparticles for Biocontrol of Citrus Blue Mold

**DOI:** 10.3390/polym17121671

**Published:** 2025-06-16

**Authors:** Mousa Abdullah Alghuthaymi

**Affiliations:** Applied College at Alquwayiyah, Shaqra University, Alquwayiyah 11971, Saudi Arabia; malghuthaymi@su.edu.sa or alghuthaymi@gmail.com

**Keywords:** biopolymers, biosynthesis, nanocomposite, orange fruits, *Penicillium italicum*

## Abstract

Citrus fruits are major economic and nutritional crops that are sometimes subjected to serious attacks by many fungal phytopathogens after harvesting. In this study, we focus on the structures of potential antifungal nanocomposites from artichoke leaf extract (Art), Art-mediated nanosilver (AgNPs), and their nanoconjugates with chitosan nanoparticles (Cht) to eradicate the blue mold fungus (*Penicillium italicum*) and preserve oranges during storage via nanocomposite-based edible coatings (ECs). The biosynthesis and conjugation of nanomaterials were verified using UV and infrared (FTIR) spectroscopy, electron microscopy (TEM and SEM) analysis, and DLS assessments. Art could effectually biosynthesize/cap AgNPs with a mean size of 10.35 nm, whereas the average size of Cht was 148.67 nm, and the particles of their nanocomposites had average diameters of 203.22 nm. All nanomaterials/composites exhibited potent antifungal action toward *P. italicum* isolates; the Cht/Art/AgNP nanocomposite was the most effectual, with an inhibition zone of 31.1 mm and a fungicidal concentration of 17.5 mg/mL, significantly exceeding the activity of other compounds and the fungicide Enilconazole (24.8 mm and 25.0 mg/mL, respectively). The microscopic imaging of *P. italicum* mycelia treated with Cht/Art/AgNP nanocomposites emphasized their action for the complete destruction of mycelia within 24 h. The orange (*Citrus sinensis)* fruit coatings, with nanomaterial-based ECs, were highly effectual for preventing blue mold development and preserved fruits for >14 days without any infestation signs; when the control infected fruits were fully covered with blue mold, the infestation remarks covered 12.4%, 5.2%, and 0% of the orange coated with Cht Art/AgNPs and Cht/Art/AgNPs. The constructed Cht/Art/AgNP nanocomposites have potential as effectual biomaterials for protecting citrus fruits from fungal deterioration and preserving their quality.

## 1. Introduction

Citrus fruits are diverse and highly important fruits, but they are frequently subjected to attack from phytopathogenic microbes and are damaged throughout the processes of farming, ripening, harvesting, or commercialization because of their elevated nutritional components and high water content [1]. The acidic nature of citrus fruits (typical pH ~2.2–4.0) could drive fungal phytopathogens to cause severe infections, leading to the deterioration of harvested fruits [2]. Contamination with/invasion of phytopathogens can occur in each stage of a plant’s life. *Penicillium italicum* is among the necrotrophic fungal phytopathogens that cause blue mold infestation (the second leading cause of postharvest decay in citrus crops); such infestations lead to massive agricultural/economic losses. Regarding the disease cycle of *P. italicum*, at 25 °C, 3–5 days is required to reproduce 1–2 billion conidial spores [1,2].

The many limitations/controversies associated with existing synthetic and chemical fungicides for managing harvested crop infestation have led to them being disused in most countries. These limitations/controversies include the persistence and accumulation of fungicides in the environment, their contribution to the development of resistant fungal strains, the health risks posed to humans/animals through direct exposure or consumption, their contribution to the contamination of soil and water sources, and their unintended effects on beneficial organisms, reducing the effectiveness of their application [1,2,3].

The frequency at which chemical/synthetic fungicides are applied has decreased globally because of their probable carcinogenicity, acute toxicity, and effects on the environment, alongside difficulties regarding plant degradation and tighter regulations. Consequently, fungi resistance has been stimulated, and there is now increased public concern regarding pesticide residues in harvests [3]. Thus, the idea of exploiting natural derivatives and biocontrol agents (e.g., antagonistic microbes, bioactive bio-derivatives, and nano-biomaterials) is receiving increasing attention because effectual, environmentally friendly, and safe biodegradable alternatives are needed to overcome postharvest infestations and ensure that the risks posed to humans and the environment are minimal [4].

*Cynara scolymus* (artichoke), a species of edible plant, is one of the most promising natural alternatives. Artichoke leaf extract (Art) could possess diverse bioactivities (e.g., antioxidant, anti-allergenic, anti-inflammatory, anti-hepatocellular carcinoma, and anti-ulcerogenic activities) [5]. Art provides a rich source of bioactive metabolites, including cynarine, chlorogenic acid, apigenin, luteolin, caffeic acid, and flavonoids [6]. Artichoke leaf extract (Art) has historically been applied in herbal remedies in experimental and clinical trials because of its promising bioactivities (e.g., its hepatoprotective, antioxidant, lipid-lowering, bile-enhancing, and choleretic effects). Art was said to have numerous beneficial effects, in particular on liver function. In vivo animal treatments with Art highlight its role in supporting and regenerating liver tissues, in addition to its cholesterol-lowering effect, helping to maintain heart functionalities [7,8]. Boiled Art can reduce insulinemic responses and postprandial glycemia with no side effects on those with metabolic syndrome [8]. The part of the artichoke that is most often consumed is the head, meaning that most of the plant mass (80–85%) is considered as biomass waste. This large amount of waste contains numerous phenolic bioactive components with high potential for the biosynthesis/reduction of nanometals (e.g., AgNPs) [5,6,7]. The inexpensive and eco-friendly nature of artichoke wastes, as well as their elevated contents of bioactive compounds (e.g., phenolic acids derived from caffeoylquinic acid or flavonoids such as luteonin and apigenin, alongside biopolymers like inulin and pectin), with strong antioxidant and reducing powers, supports their targeted usage as reducing agents to prepare AgNPs [6,7].

Nanotechnology and nanoparticles (NPs) are currently employed in almost all human-related fields, having biological, nutritional, biochemical, biomedical, mechanical, optical, agricultural, and environmental applications [9]. NPs have been integrated into food production, manufacturing, and preservation chains, from farming and fertilization to presentation for consumption [10]. NPs have unique traits, including their minute size, very large surface area, excellent penetrability, and unique shapes and distributions, meaning that they possess enhanced and novel characteristics not found in bulk particles. In terms of physical methods, the typical drawbacks associated with NP synthesis protocols include the inherently excessive energy requirements and elevated costs, as well as limited NP yields. Meanwhile, in chemical procedures, the main drawbacks are the serious toxicological and ecological consequences of their use [11]. The use of biosynthesis-derived (i.e., green synthesis-derived) biomaterials for NP synthesis (e.g., plant materials/derivatives, microbes, biopolymers, algae, or other biomolecules) can help to effectively manage most of the aforementioned drawbacks, facilitating environmentally friendly, easy-to-apply, cost-effective, high-yield-producing, and controllable approaches [12]. Plant (phyto) extracts and other phytochemicals have been effectively exploited to synthesize metal NPs using “green approaches”; numerous phytochemicals (e.g., flavonoids, alkaloids, phenolics, carotenoids, lignans, terpenoids, and other bioactive molecules) have been authorized for use to mediate, stabilize, and cap NPs with astounding traits [11,12].

Silver nanoparticles (AgNPs) in particular are attracting increasing interest among metal NP researchers due to their distinct biological, physicochemical, and antimicrobial features, supporting the diverse application of AgNPs in numerous fields (e.g., fields related to electrochemical and optical sensors, pharmaceuticals, environmental research, and food production). The extensive uses of AgNPs are mostly based on their outstanding luminescent traits, easy-to-implement synthesis approaches, heightened bioactivity and sensitivity, elevated chemical stability, and measurement simplicity [13]. The biosynthesis of AgNPs, as well as conjugation/capping inside coated biomolecules, is strongly advised to ensure much safer NPs with minimal biotoxicity toward mammalian cells [14].

Chitosan is a positively charged deacetylated biopolymer derived from chitin, which has elevated bioactivities and functionalities [15]. It has been reported that the bioactive physiognomies of chitosan (as a biodegradable, nontoxic, and eco-friendly biopolymer) have the potential to exert bactericidal, wound-healing, antioxidant, biosorption, and anticancer effects [15,16]. Chitosan particles are usually transformed into membranes, emulsions, bandages, hydrogels, or edible coating (EC) films [16]. The functionalities, bioactivities, and formulability of this charged biopolymer can be greatly augmented by transforming it into chitosan nanoparticles (Cht), which are appropriate for individual use or as functional carriers for additional biomolecules in areas related to biomedicine, pharmaceuticals, the environment, agriculture, nutrition, and food processing [16,17]. Cht is currently the ideal nanopolymer for fabricating bioactive functional ECs to protect foods/crops from microbiological spoilage, oxidative stress, and pathogen invasion and, therefore, protect plant quality [17,18]. It has been stated that the use of Cht for capping/stabilizing further nanomaterials (e.g., nanometals such as silver and selenium, nanopolymers, and phytocompounds) can reinforce their bioactivities, functionalities, and applicability as antimicrobial, health-promoting, and tissue-guarding agents [15,16,17]. The specific innovations of the current study include the fact that we used Art, a readily available and cost-effective bioproduct, to synthesize AgNPs and strengthen their antifungal powers, employing Cht as a stabilizing and capping agent to facilitate their attachment/dispersion and reduce their potential toxicity. This was performed to provide effective alternatives derived from natural sources to address the problem of hazardous fungicides [2,5,10,15,17].

In this investigation, we aimed to innovatively synthesize and produce nanocomposites from Cht and biosynthesized AgNPs using Art to construct bioactive nanocomposites and assess their effectiveness as *P. italicum* fungicides. Their ECs were applied to control blue mold development in oranges.

## 2. Materials and Methods

### 2.1. Art (Artichoke Extract) Preparation

Organically farmed artichoke (*Cynara scolymus*) was acquired from an ISO-certified market in Jeddah, Saudi Arabia. Artichoke leaves were detached manually from the plant pulp and head, before being cleansed repeatedly with DDW (double-distilled water), NaOCl solution (5% concentration, 4 min), and DDW again. Then, fresh leaves were dried (using an air oven at 48 ± 2 °C) for 40–48 h. The dried leaves were mechanically pulverized, and the powder was extracted via immersion in ethanol (70%, 10 folds, *w*/*v*), before being rotated in a shaker (KS-4000 I control, IKA, Staufen, Germany) at 122× *g* for 35 h at RT “room temperature; 25 ± 2 °C”, prior to filtration to exclude leaf residues. Art was vacuum-dehydrated (R100, Rotavap; Büchi, Switzerland) at 42 °C, and then, the obtained materials were re-dissolved in DDW to acquire a 10% (*w*/*v*) concentration.

### 2.2. Art-Based Biosynthesis of AgNPs

A freshly prepared stock solution (0.01 M) of silver nitrate (AgNO_3_), Sigma-Aldrich, St. Louis, MO, USA) was produced in DDW. Consequently, equal volumes (15 mL) of both Art (1.0%, *w*/*v*) and AgNO_3_ solutions were intermixed under vigorous stirring (640× *g* in VELP Scientifica Srl., Via Stazione, Italy) for 150 min at RT. The Art-based biosynthesis of AgNPs was confirmed when the mixture solution turned dark brown. Art/AgNPs were harvested via centrifugation-based precipitation (SIGMA 2–16 KL; SIGMA Laborzentrifugen GmbH, Osterode am Harz, Germany) for 38 min at 11,450× *g*. To obtain plain AgNPs, the Art/AgNPs were washed (4 times) with DDW and, subsequently, with ethanol. They were centrifuged after each wash at 11,450× *g* for 35 min [18,19]. Both the Art/AgNPs and plain AgNPs were subsequently freeze-dried.

### 2.3. Nanocomposite (NC) Preparation

The preparation/loading of Cht with Art/AgNPs followed previously described protocols [20,21,22]. Cht (medium molecular weight; Sigma-Aldrich, St. Louis, MO, USA; Product Number: 448,877, deacetylation > 75%) was employed for NC production. TPP (5-Sodium tripolyphosphate; Sigma-Aldrich, MO, USA) was used to cross-link Cht nanoparticles. Cht (0.1%, *w*/*v*) was dissolved in diluted acetic acid (1.5%, *v*/*v*); then, a half-volume of TPP solution (0.6%, *w*/*v* in DDW) was dropped gently (with 0.28 mL/min rate) into Cht solution with vigorous stirring (685× *g*). Stirring was sustained for a further 70 min, following TPP addition, and then the developed Cht-TPP nanomaterials were gathered by centrifugation (11,450× *g*). For conjugated NC formation from Art/AgNPs and Cht (Cht/Art/AgNPs), the Art/AgNP solution (0.1% *w*/*v*, in DDW) was gently amended to a Cht solution at equal amounts and vigorously vortexed (685× *g* for 115 min) prior to the addition of TPP into a mixed solution. Subsequently, the nanoparticles/NCs were centrifuged, washed with DDW, and freeze-dried.

### 2.4. Nanomaterial Characterization

#### 2.4.1. FTIR Spectroscopy

The employed compounds (Art, Cht, Art/AgNPs, and Cht/Art/AgNPs) were analyzed spectrophotometerically after mixing with KBr (at 1%, *w*/*w*) via FTIR (FT-IR-360, Fourier transform infrared spectroscopy, JASCO, Tokyo, Japan) within the 450–4000 cm^−1^ wavenumber range in transmission mode.

#### 2.4.2. Assessment of Particle Charges and Sizes

The charges (zeta potential; ζ) and sizes (Ps) of Cht and Art-synthesized AgNP particles, as well as of their NCs (Cht/Atr/AgNPs), were assessed via a DLS (dynamic light scattering) approach involving zetasizer (Brookhaven; Zeta plus; Nashua, NH, USA).

#### 2.4.3. Ultrastructure of Nanoparticles

The Art-biosynthesized AgNPs’ shapes, sizes, and distributions were visualized via TEM (transmission electron microscopy, Leo 0430; Leica, Cambridge, UK) imaging after they were mounted onto TEM carbon grids. SEM imaging (Scanning electron microscopy; JEOL; JSM-IT100; Tokyo, Japan) reflected the Cht/Art/AgNPs’ features and topography; the NCs were dehydrated using gradual ethanol, coated (gold/palladium), and then screened at an acceleration value of 20 kV.

### 2.5. Blue Mold Isolates

*Penicillium italicum* isolates (e.g., *Pi* A—standard *Penicillium italicum* Wehmer strain; ATCC-48955; *Pi* O—isolated from orange; and *Pi* L—isolated from lemon) were attained and identified at the Department of Plant Protection, King Saud University, KSA, Riyadh, Saudi Arabia. The identities of isolated fungi were further proved using MALDI-ToF MS (Matrix-assisted laser desorption/ionization adjoining time-of-flight mass spectrometry; Bruker, Billerica, MA, USA analysis). All fungal isolates were regularly grown and challenged via consumption with PDB/PDA media (potato dextrose broth and potato dextrose agar; Oxoid, UK) at 27 ± 1 °C aerobically. Fungal SSs (spore suspensions) were formulated through the gentle scraping of 7-day-developed fungi on PDA with a scraper. Spores were amended in DDW, vortexed, counted (via an automatic cell counter; Thermo Fisher Scientific, Countess, Waltham, MA, USA), and then their final count was adjusted to ~2.5 × 10^6^ spores/mL with DDW.

### 2.6. Evaluation of Blue Mold Antifungal Activity In Vitro

The blue mold fungicidal actions of Cht, Art/AgNPs, and Cht/Art/AgNPs were assessed in vitro for their potential to inhibit fungal isolates. Enilconazole (syn. Imazalil; Sigma-Aldrich, Steinheim, Germany) was employed as a comparative fungicide in efficacy assessments. It was dissolved in DMSO (dimethyl sulfoxide, Sigma-Aldrich, Germany, 20% *v*/*v*).

#### 2.6.1. Qualitative WD (Well Diffusion) Method

The qualitative WD technique was employed to appraise the antifungal potential of the screened materials (Cht, Art/AgNPs, and Cht/Art/AgNPs) [23]. Wells with an average diameter of 6 mm were made using a cork borer on inoculated PDA plates with 100 µL of mold SS. Portions of 50 µL (comprising 1.0% concentration from each material in DDW or enilconazole in DMSO) were dropped into the plates’ wells. After the plates were subjected to incubation in the dark for up to 75 h at 27 ± 2 °C, the emerged IZs (inhibition zones) surrounding the wells were measured in millimeters [18].

#### 2.6.2. Quantitative MFCs (Minimum Fungicidal Concentrations)

The MFCs of the screened agents (Cht, Art, Art/AgNPs, and Cht/Art/AgNPs) or enilconazole toward *P. italicum* isolates were evaluated via a diluted broth-based protocol [24]. Each compound was intermixed with PDB in their respective concentrations (from 10 to 100 mg/mL) and inoculated with fungal SS. The broths were incubated aerobically (as above) for 4 days; then, 0.1 mL of each broth was spread over fresh PDA and further incubated for 4 days. PDA plates without any apparent mycelia after incubation indicated the MFC of each agent against the aforementioned isolates. Enilconazole was used as a standard fungicide for comparison.

### 2.7. Orange Edible Coating (EC) with Antifungal Nanocomposites

#### 2.7.1. EC Preparation

ECs were prepared following the procedure described by Salem et al. [18]. Briefly, the nanomaterials (Cht, Art, Art/AgNPs, and Cht/Art/AgNPs) were liquefied in DDW and acidified to pH 5 (at corresponding MFC levels); the integrated plasticizer was glycerol (at 5%, *v*/*v*).

#### 2.7.2. Orange Treatment

The oranges studied (*Citrus sinensis*, var. Navel, organically farmed) were produced in Egypt, cooled, and delivered to the lab free of injuries, superficial damage, or signs of infection; the average fruit diameter was 7.6 ± 0.6 cm. The fruits were sanitized via immersion in NaOCl (5%) solution and washing with DDW prior to coating. After the fruits were wounded with a sterile cutter (2.5 mm deep × 3 mm wide) at a position in their middle, the wounded fruits were flooded with a 1200 mL solution containing a fungal SS of *Pi*O for 6 min (to promote simulated infections). The fruits were drained (16 min), air-dried (95 min), and then dipped into EC solutions (~1 L) for 7 min at RT. Then they were drained, air-dried, and preserved in polyethylene containers. The fruits coated with acetic dilutions represent group C (negative control group). Plain AgNPs were not selected as negative controls because it is recommended that plain AgNPs are excluded from direct food applications due to their biotoxicity [13,14]. The EC-treated oranges were retained in a humidified room (sterile; 89% RH; at RT) for 2 weeks. The diameters of fungal decay features (LDs) were measured routinely throughout the inoculation period [25].

#### 2.7.3. Microscopic Optical Observations of Treated Mycelia

After exposure to MFCs from Cht/Art/AgNPs, signs of *P. italicum* fungal growth and the topography were screened microscopically (Labo America Inc., Labomed Lx400; Fremont, CA, USA). Fungal mycelia were stirred and incubated with nanocomposite MFCs for 24 and 48 h. Then, treated mycelia were stained with lactophenol blue (Sigma-Aldrich, MO, USA) and subjected to microscopic imaging.

### 2.8. Statistical Analysis

Triplicated experiments were analyzed using SPSS V-20 software to compute the means and SDs (standard deviations). To compare mean values and significant differences (*p* ≤ 0.05), a one-way ANOVA test was employed.

## 3. Results and Discussion

### 3.1. Appearance of Art-Biosynthesized AgNPs

The biosynthesis of AgNPs with Art was achieved within 90 min of interaction, signified by the mixture solution changing color from clear to blackish-brown, following the conjugation of AgNO_3_ with Art (Figure 1A). The noticeable color change in the AgNO_3_ and Art solutions verified the reduction of Ag^+^ to Ag^0^ (AgNPs), liberating electrons and conferring SPR (surface plasmon resonance) absorption excitation. The absorption UV-vis spectrum of the Art-synthesized AgNPs showed a distinctive maximum peak at 431 nm (Figure 1B), belonging to the SPR-designated range for biosynthesized AgNPs (within 220–450 nm) [26,27,28,29,30]. These reports also suggested a relation between the frequency/width of the SPR absorption band and AgNP size, dielectric constant, and shape. This sole SPR band indicates that the AgNPs have a spherical shape, whereas the ≥2 distinctive SPR bands are often related to anisotropic NPs [28,29,30]. The pattern of UV-vis spectroscopy for the Art/AgNPs can be used to identify the size-controlled particles biosynthesized using Art, indicating the extract’s phytosynthesis efficiency and capability to reduce/cap Ag^+^ to AgNPs [26,28,29].

### 3.2. FTIR Analysis

The biomolecules in Art responsible for the reduction/stabilization of AgNPs and the results of Cht/Art/AgNP conjugation were evaluated via FTIR analysis (Figure 2). The key groups/bonds in Art responsible for AgNP biosynthesis are illustrated in Figure 2-Art. The IR peak around 3248.4 cm^−^^1^ corresponds to O–H vibration stretches (of phenols, water, carbohydrates, alcohols, and peroxides), while the band at 2929.1 cm^−1^ belongs to the C–H vibration stretches of CH_3_/CH_2_ in lipids and aldehydes. These compounds can mostly facilitate the reduction of Ag^+^ to Ag^0^. The IR peak at 1573.9 cm^−1^ is linked to a C–C aromatic ring, whereas the 1383.2 cm^−1^ peak corresponds to C–O–C stretches [19,31]. The 1603.7 cm^−1^ peak relates to the aromatic domain and N–H vibration and bending in amino acids, whereas the observed band at 1397.8 cm^−1^ reflects the vibrated stretching of C–O in amides and the C–C stretches of phenyl groups. The weak 1334.2 cm^−1^ peak indicates the stretched (C–N) frequency and indicates the existence of amine groups. The 913.6 cm^−1^ peak points towards the bending (=C–H) frequency in alkene groups, whereas the stretching vibrations of carbonyl (C–O or O–H) bending are detected at the 1254.6 cm^−1^ band [32]. Stretching C–O or C–O–C linkages (of phenolic compounds) are indicated by the intense peak at 1018.5 cm^−1^, mainly associated with cynarin and further flavonoids that exist in *C. scolymus* extract [5,32]. The peak located at 671.6 cm^−1^ relates to the vibration of the –CH_3_ group and C–alkyl chloride.

The Art phytomolecules that possibly contributed to AgNP biosynthesis/capping are clearly evident based on the comparison between the plain Art spectrum and Art/AgNPs’ spectrum (Figure 2-Art/AgNPs). The disappearance of many functional carbonyl and hydroxyl groups indicate their roles in AgNP bioreduction [5,31,32]. Numerous peaks (in the Art spectrum) disappeared due to the biosynthesis/capping of AgNPs, as evident in their combined spectrum (Figure 2-Art/AgNPs). The missing bands are indicated by red zones, indicative of Art biogroups after interaction with AgNPs. Further novel peaks emerged/sharpened after AgNP interactions with Art (highlighted with blue lines), indicating the generation of new bonds between AgNPs and Art biomolecules.

The IR spectrum regarding Cht (Figure 2-Cht) revealed biopolymers indicative of biogroups/bonds, including the occurrence of amide III, amide I, and the stretched N–H/O–H, at wavelengths of 1421.3, 1633.5, and 3374.4 cm^−1^, respectively, whereas the 1031.3 cm^−1^ band represents C–C and C–O stretching. Typical bands in the Cht spectrum corresponding to native chitosan were identified at wavenumbers of 1073.8 cm^−1^ (stretched C–O glycosidic bonds), 1139.7 cm^−1^ (C–O–C bonds), 1374.9 cm^−1^ (bridge O stretch), 1613.6 cm^−1^ (N–H bending), and 2880.4 cm^−1^ (stretched C–H) [21,33,34].

The conjugated spectrum of Cht/Art/AgNPs reflected the existence of numerous peaks that had been transferred from Cht (highlighted with green spots), in conjugation with extra peaks from Art/AgNPs (Figure 2. Cht/Art/AgNPs). The occurrence of diverse peaks from both conjugated biomolecules validated their biochemical and electrostatic interactions, as reported in previous investigations studying Cht-based nanocomposites [35,36,37,38,39].

### 3.3. Structural Analysis of Nanomaterials

The phytoreduction of silver nitrate (AgNO_3_) to AgNPs using Art was demonstrated visually by the solution changing color from clear to intense blackish-brown. The ζ potential and size values of Cht, Art/AgNPs, and Cht/Art/AgNPs are shown in Table 1.

Art could reduce AgNPs to tiny nanoparticles with an average Ps of 10.35 nm in diameter. The Art-biosynthesized AgNPs exhibited elevated negative charges (ζ potential = −23.8 mV), which helped in preserving AgNP dispersion and avoiding their aggregation. For Cht, the nano-biopolymer average Ps was 148.67 nm in diameter, with high surface positivity (+37.5 mV). After nanocompositing the nanoparticles (to prepare Cht/Art/AgNPs), their particles had bigger Ps values, with average diameters of 203.22 nm, indicating that the Art/AgNPs were integrated with/capped within Cht molecules. Moreover, the Cht/Art/AgNP nanocomposites were less positively charged (+30.9) than plain Cht, which suggests that the Art/AgNPs were copiously embedded in biopolymer nanoparticles, where their external surfaces comprised Cht.

Electron microscopy was able to accurately capture nanoparticle shapes and dissemination (Figure 3). The carried charges (ζ potential) of the constructed nanoparticles/composites capably conserved their stability, especially for the Art/AgNPs (Figure 3A). The TEM picture of phytosynthesized AgNPs, captured by Art topography (Figure 3A), signified the diminished Ps of AgNPs and their homogenous spherical size/distribution (within a 3.16–27.49 nm range and a 10.35 nm diameter mean) that match the DLS analysis results (Table 1). The SEM ultrastructure image of the Cht/Art/AgNP nanocomposites reflects their irregular/semispherical shapes (Figure 3B); their average Ps was ~205.71 nm, and a good distribution was observed. The phytoconstituents in Art (mainly from phenolics compounds and flavonoids) can act as reducing/stabilizing agents for the reduction of AgNPs [40,41,42].

The high competency of Cht with respect to capping diverse nanometals/extracts and creating vast constant nanocomposites with a very low Ps was formerly verified [33,38]. NPs/nanocomposites with higher ζ potentials (≤−30 mV or ≥+30 mV) mostly display outstanding stability and dispersion patterns among particles, resulting from electrostatic repulsion [28,43].

The effectiveness of the Cht nanoparticle synthesis protocol was verified through TPP cross-linking, facilitating an ionic gelation-based interface; Cht synthesis via this procedure provides applicable nanomaterials, as basic bioactive biopolymers, as nanocarriers for extra biomolecular constituents, and as bases to construct bioactive ECs [18,21,37].

### 3.4. Nanomaterials’ Inhibition Activity Against Penicillium Italicum

In vitro assessments of *P. italicum* inhibition among the investigated agents (Cht, Art, Art/AgNPs, and Cht/Art/AgNPs), along with enilconazole as a comparative fungicide, were carried out through qualitative/quantitative assaying approaches (Table 2). All agents/nanocomposites showed significant fungicidal activity against *P. digitatum* isolates. The Cht/Art/AgNP nanocomposites exhibited the strongest action, with significant activity exceeding that of the other compounds and the enilconazole fungicide. Synergistic antifungal action was demonstrated among the screened agents, and the conjugation of various agents (Art/AgNPs and Cht/Art/AgNPs) exhibited additional forceful consequences compared to the distinct compounds (namely Cht and Art).

Regarding the sensitivity of *P. italicum* isolates to challenging materials, the *Pi* L isolate exhibited the highest resistance, whereas *Pi* A had the highest sensitivity, evidenced by the MFC and ZOI values (Table 2). All *P. italicum* strains exhibited “high susceptibility” to both Art/AgNPs and Cht/Art/AgNPs, and they were considered as “moderately susceptible” toward Cht and Art.

The antimicrobial (especially antifungal) potential of the investigated agents was evaluated against diverse phytopathogens. Cht’s microbicidal traits principally involved positive surface charges, facilitating attachment/adhesion and its interaction with microbial cells/membranes and inner organelles. An increase in intracellular ROS (reactive oxygen species) production, as well as an upsurge in the permeability of membranes and the suppression of microbial bioactivities, has been reported after Cht exposure [44,45]. Cht’s transformation to nanoforms could augment these bioactivities due to the amplified surface reactions and minute Ps, which enable added powerful actions and biocidal interactions [33,36,44].

For the innovatively conjugated Cht/Art/AgNPs in this study, forceful synergistic fungicidal actions stemming from the compositing materials (Cht, Art, and AgNPs) were verified in terms of their minimum MFC and broadest ZOI values; this emphasizes the potential of nanocomposite ingredients to substantiate the combined antifungal mechanisms [18,22,39]. Moreover, using Cht to carry, cap, and deliver bioactive materials/compounds (e.g., nanometals, extracts, and phytochemicals) was found to amplify their joint actions as antibiotics or anticancerous or antioxidant nanocomposites [35,36,37,38], which supports the value of Cht in reinforcing the fungicidal actions of both Art and AgNPs.

Cht’s fungicidal action modes are still not well understood, but we aimed to include interactions between the positively charged polymers with hyphal walls, interaction with mycelial membranes, and penetration through such membranes to destroy/hinder fungal biosystems [18].

The antifungal (and general antimicrobial) potential of Art was proven via testing on diverse fungi, yeast, and bacteria; the bioactive constituents in Art (mainly from high phenolic and flavonoid contents) prompted its activity against microorganisms [46,47,48]. The main phenolic phytocompounds isolated from Art included cynarin, chlorogenic acid, four derivatives of caffeoylquinic acid, and 3,5- and 4,5-di-*O*-caffeoylquinic acid, whereas the key flavonoids comprised cynaroside, luteolin-7-rutinoside, apigenin-7-*O*-β-D-glucopyranoside, and apigenin-7-rutinoside [47,49]. In particular, the phytocompounds cynarin, chlorogenic acid, cymaroside, and luteolin-7-rutinoside exhibited outstanding activities against microorganisms, especially against fungi, recording an antifungal MFC range of 50–200 μg/mL [47,50]. The Art bioactive phenolics and flavonoids also possess noticeable antioxidant and anticancer activities, in addition to their antimicrobial actions [48]. The main antifungal compounds identified in Art included di-O- and mono-O-caffeoylquinic acids, in addition to sesquiterpene cynaropicrin, pointing toward the phytopathogenic *Alternaria alternata* fungus [48,50].

### 3.5. Microscopic Observations of Blue Mold Treated with Nanocomposites

The distortions/variations in the morphology of *P. italicum* mycelia after Cht/Art/AgNP-based treatment were screened microscopically, and MFC values were noted (Figure 4). The *P. italicum* mycelia at the start of the trial had healthy and rigid walls. Their walls/surfaces were dense and uniform with no superficial distortions (Figure 4A). After contact for 24 h, the mycelia were irregularly swollen and fragmented, with remarkable signs of distortion and wall softening (Figure 4B). At the expiration of exposure, after 48 h, the *P. italicum* mycelia were entirely lysed and mostly devoid of their characteristic structures; the intracellular matrix had leaked completely from their hyphae at this stage (Figure 4C).

The biocidal action of AgNPs (at a 75 ppm concentration) has been observed to effectively the inhibit spore germination and mycelial growth of diverse plant rot pathogens (e.g., *Alternaria alternata*, *Botryosphaeria dothidea*, *Diaporthe actinidiae*, and *Pestalotiopsis microspora*) via increasing mycelial membrane permeability, leading to intracellular substance leakages [51]. Electron microscopy observations (SEM and TEM) revealed severe hyphal distortion, vacuolation, and shrinkage after fungal treatments with AgNPs, reflecting the organelle- and cell-based structural degradation caused by the AgNPs [14,28,51]. Biogenic AgNPs are also said to possess remarkable antifungal action against *A. alternata* (at 150 ppm; average particle size = 10.0 nm) and *Diaporthe* sp. (at 180 ppm; average particle size = 52.0 nm) [14,51]. Additionally, the required MIC for AgNPs/Cht to inhibit *Botrytis cinerea* was μg/mL [52].

The antifungal actions of Cht/Art/AgNPs in disrupting fungal hyphae could involve the attachment of Cht particles (positively charged) onto the mycelia, damaging the cells’ walls, affecting the penetration of nanocomposites and their components (Cht, AgNPs, and Art) inside the cells, and impacting the generation of ROS inside cells, obstruction/interaction with the interior organelles of fungi, and suppressing their functionalities, leading to fungal death and mycelial lyses [39,44,45,51].

### 3.6. Cht-Based Edible Coating for Treatment of Oranges

Biomolecule-based edible coatings of oranges (i.e., plain Cht, Art/AgNPs, and Cht/Art/AgNPs), after 14 days of infection with *P. italicum* O, were screened photographically (Figure 5). The uncoated fruits (control) were mostly decayed by blue mold and covered with growing fungi, while the coating-treated fruits displayed signs of infection (Figure 5). Infestation was entirely prevented in fruits coated with Cht/Art/AgNP-based ECs, preserving their fresh look and maintaining their overall quality. The infestation remarks on the Cht-coated orange covered ~12.4 ± 1.6% of the fruits’ surfaces, whereas in the Art/AgNP-treated fruits, *P. italicum* infestation covered ~5.2 ± 0.8% of the fruits’ surfaces (Figure 5). Interestingly, the quality of the oranges coated with Cht/Art/AgNP nanocomposites (at 1.0 and 1.5 × MFC) was sustained for an additional 10 days after the trial period, with no infestation remarks being noted. As the infestation of orange by blue mold is mainly superficial and visible on the outer peel, photographs of the oranges’ surfaces, rather than the internal tissues, were captured to provide more reliable evidence of the features of the infected fruits.

The incorporation of the AgNPs discussed herein within biomolecules (e.g., Art and Cht) is believed to substantially diminish their potential biotoxicity, as the conjugation of nanometals in biopolymer capsules has been observed to reduce their toxicity while increasing their combined bioactivity [52,53,54]. Promisingly, the application of AgNPs was confirmed to significantly reduce infestation symptoms on fruits without any Ag^+^ residues on their peels or flesh [51,53]. Ag ions and clusters (Ag, Ag^2^, and Ag^3^) could accumulate in the fungal mycelium, where the mono-Ag ions adduct with phenylglycine amino acids, Coenzyme A nucleotides, LeuSerAlaLeuGlu peptides, and diverse lipid and phospholipid derivatives [51,55]. Microscopic investigations of fungal ultrastructures after AgNP exposure revealed remarkable alterations (e.g., ultrastructural reorganization, hyphal shortening/condensation, cell plasmolysis, increased vacuolization, membranous structures, cytoplasm collapse, lipid material accumulation, mitochondrial condensation, organelle disintegration, nuclear deformation, apoptotic body formation, and chromatin condensation/fragmentation [55,56]).

Further studies have indicated that AgNPs have direct action in inhibiting spore germination and fungal mycelia growth, in addition to effects on cell membrane composition and the synthesis of the proteins, sugars, and N-acetylglucosamine in fungal cells [51,55,57,58]. Cht-based ECs have been effectively employed to preserve diverse crops and fruits, whether as a sole component or as conjugates/carriers for other bioactive molecules/nanomaterials [59,60]. The main functions of Cht in protecting harvested crops and formulating functional ECs involve the biopolymer’s potential to serve as an effectual antimicrobial molecule (exerting antifungal, antibacterial, and insecticidal activity, among other activities) due to its positive charge, synergistic functions that help strengthen the biocidal and antioxidant powers of added biomolecules, and barrier-forming properties that protect crops from oxidation and pathogen invasion [54,61].

## 4. Conclusions

In light of the growing demand for effectual and eco-friendly approaches to controlling blue mold infestation in citrus fruits, AgNPs were biosynthesized with Art and nanocomposited with Cht to generate bioactive antifungal ECs. The biosynthesis of AgNPs with Art generated tiny particles (mean size = 10.35 nm), whereas the Cht/Art/AgNP nanocomposites had a mean diameter of 203.22 nm. The produced nanomaterials/composites showed elevated antifungal action against *Penicillium italicum*; the Cht/Art/AgNP nanocomposites had the strongest action. Blue mold infestation remarks were entirely prevented for ≥24 days on oranges coated with Cht/Art/AgNPs. The fabricated Cht/Art/AgNP nanocomposites could hold promise as effectual biomaterials for protecting citrus fruits from fungal infestation and maintaining their quality. The constructed nanocomposites are cost-effective and have outstanding particle stability, and they also have diminished cytotoxicity, meaning that, with their use, there is a lower risk of diminished fruit quality due to the green synthesis approach used to prepare them, featuring encapsulation within a Cht biopolymer.

## Figures and Tables

**Figure 1 polymers-17-01671-f001:**
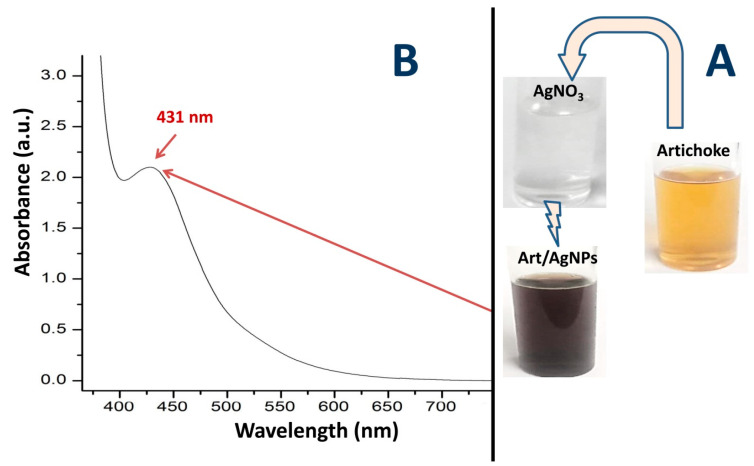
Visual color change (**A**) and UV spectral analysis (**B**) of biosynthesized AgNPs with artichoke leaf extract.

**Figure 2 polymers-17-01671-f002:**
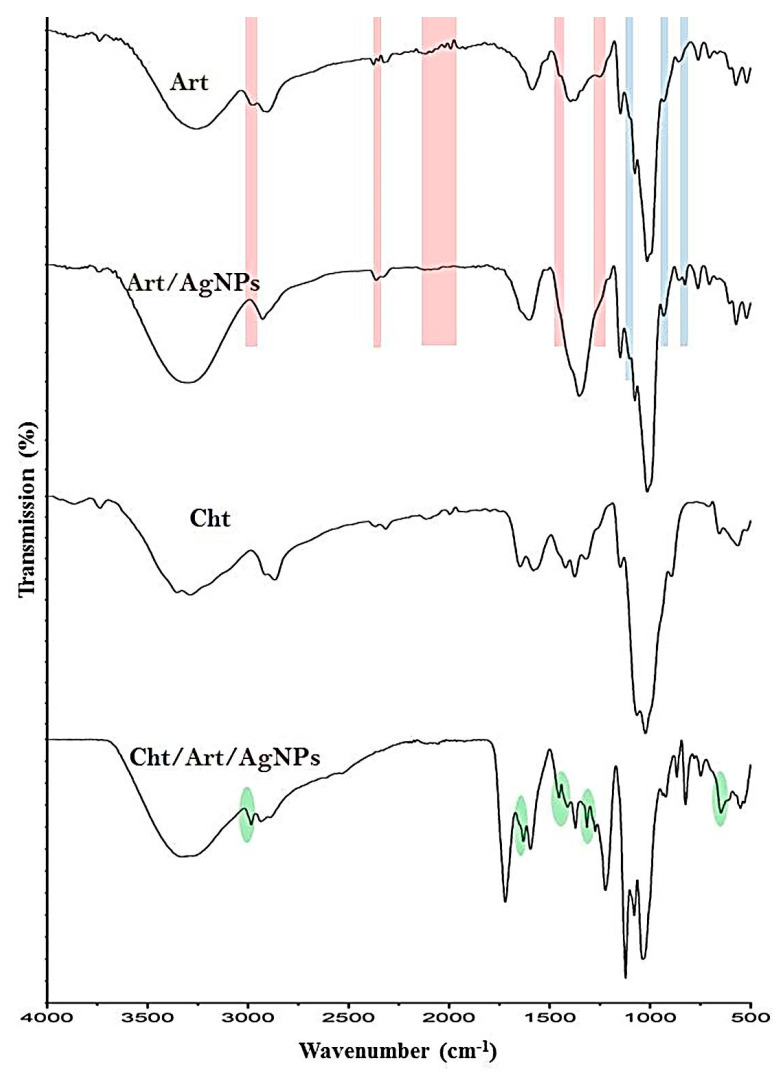
Infrared (FTIR) spectra of artichoke extract (Art), Art-synthesized AgNPs, chitosan (Cht), and their nanoconjugates (Cht/Art/AgNPs).

**Figure 3 polymers-17-01671-f003:**
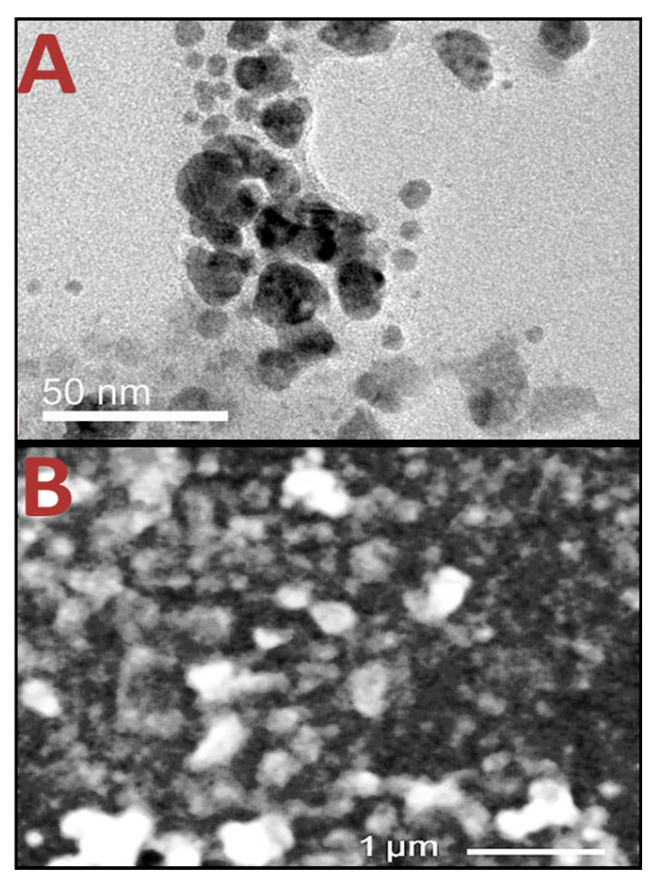
Electron microscopy features of nanomaterials, including TEM image (**A**) of artichoke extract-mediated AgNPs and SEM image (**B**) of chitosan/artichoke extract/AgNP nanocomposites.

**Figure 4 polymers-17-01671-f004:**
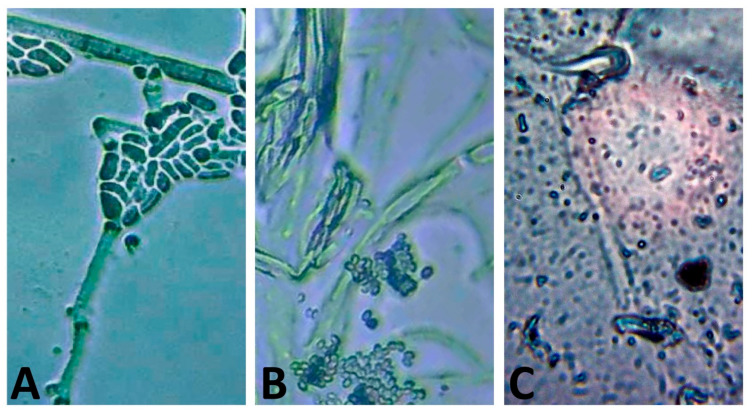
Visual microscopic screening of *Penicillium italicum* mycelia exposed to nanocomposited chitosan/artichoke extract/AgNPs for 0 h (**A**), 24 h (**B**), and 48 h (**C**).

**Figure 5 polymers-17-01671-f005:**
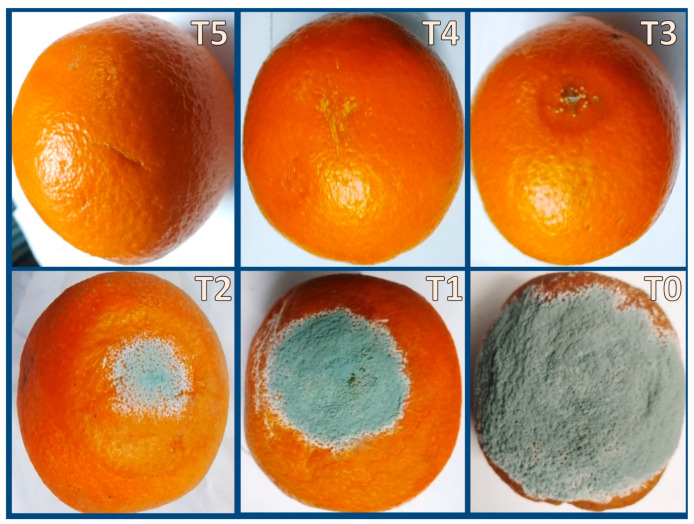
Development of blue mold (*Penicillium italicum*) infestation in oranges coated with nanocomposites after 14 days of incubation. T0: control (acetic solution); T1: artichoke extract; T2: chitosan; T3: artichoke/AgNPs; T4: chitosan/artichoke/AgNPs (1.0 × MFC); T5: chitosan/artichoke/AgNPs (1.5 × MFC).

**Table 1 polymers-17-01671-t001:** Zeta (ζ) potential and Ps distribution of nanochitosan (Cht) artichoke extract-synthesized AgNPs (Art/AgNPs) and their nanoconjugates (Cht/Art/AgNPs).

Nanoparticle	Size Range (nm)	Mean Diameter (nm)	ζ Potential (mV)
Cht	46.72–392.51	148.67	+37.5
Art/AgNPs	3.16–27.49	10.35	−23.8
Cht/Art/AgNPs	69.44–492.15	203.22	+30.9

**Table 2 polymers-17-01671-t002:** Results derived from in vitro assessments of antifungal action against *Penicillium italicum* among investigated materials *, including ZOI (zone of inhibition) diameters (mm) and MFC (minimal fungicidal concentration) values (mg/mL).

Antifungal Compound	*Penicillium italicum* Isolates					
	*Pi* A		*Pi* L		*Pi* O	
	ZOI ^**^	MFC	ZOI	MFC	ZOI	MFC
Cht	19.3 ± 1.4 ^a^	35.0	17.7 ± 1.7 ^a^	42.5	18.9 ± 1.8 ^a^	37.5
Art	18.2 ± 2.1 ^a^	37.5	15.8 ± 1.6 ^b^	37.5	17.2 ± 1.6 ^b^	35.0
Art/AgNPs	24.2 ± 1.8 ^b^	27.5	22.5 ± 2.2 ^c^	32.5	23.3 ± 2.5 ^c^	30.0
Cht/Art/AgNPs	31.1 ± 2.3 ^c^	17.5	27.4 ± 2.6 ^d^	22.5	28.6 ± 2.3 ^d^	20.0
Enilconazole	24.8 ± 2.2 ^b^	25.0	22.3 ± 2.0 ^c^	32.5	24.1 ± 1.8 ^c^	30.0

* The investigated components comprised nanochitosan (Cht), artichoke (Art), Art-synthesized AgNPs (Art/AgNPs), and their nanoconjugates (Cht/Art/AgNPs), and a standard enilconazole fungicide was used for comparison. ** Means from triplicate experiments are provided; different superscripts within the same column signify significant differences (*p* > 0.05).

## Data Availability

The data presented in this study are available on request from the corresponding author.

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
