# Peer review of "Antifungal Action of Edible Coating Comprising Artichoke-Mediated Nanosilver and Chitosan Nanoparticles for Biocontrol of Citrus Blue Mold"

_polymers, 2025, doi:10.3390/polym17121671_

Round 1
Reviewer 1 Report
Comments and Suggestions for Authors
Please refer attachment.

Author Response
The modified/corrected parts according to Reviewer (1) comments are highlighted with Yellow throughout the manuscript
Responses to Reviewer 1 Comments
Major Comments
- Explicitly state the % reduction in mold growth on coated oranges vs control in abstract.
Response: Done, the % reduction in mold growth on coated oranges vs control was provided in abstract
- Introduction: Authors should strengthen the rationale for using Art/AgNPs/Cht by citing more recent studies on their combined antifungal efficacy. Further, please clearly define the limitations of existing fungicides and how this study’s approach overcomes them.
Response: Done, the rationales for using Art/AgNPs/Cht on their combined antifungal efficacy was provided. Also, the limitations of existing fungicides and how this study’s approach overcomes them were clearly defined
- Methods: Should provide exact details (e.g., centrifugation time/temperature for AgNPs, Art extraction protocol). Authors should clarify if negative controls (e.g., AgNO3) were included to validate Art-AgNPs role in antifungal activity.
Response: Done, the exact details (e.g., centrifugation time/temperature for AgNPs, Art extraction protocol) were provided. Also, the use negative controls were highlighted in antifungal activity methodology.
- Results: Please elaborate on how Cht/Art/AgNPs disrupt fungal hyphae (e.g., ROS generation, membrane damage). Furthermore, compare the nanocomposite’s performance with other green-synthesized AgNPs in literature.
Response: Done, the mechanisms on how Cht/Art/AgNPs disrupt fungal hyphae (e.g., ROS generation, membrane damage) were elaborated. Additionally, results of other former silver nanoparticle-based approaches for anti-infection applications were provided in the discussion.
- Conclusion: Address practical challenges (e.g., cost, stability of coatings during storage/transport). Additionally, briefly discuss safety evaluations (e.g., potential residues on fruit, mammalian cell toxicity).
Response: Done, practical challenges (e.g., cost, stability of coatings during storage/transport) were addressed in conclusion, with brief appointment of safety evaluations (e.g., potential residues on fruit, mammalian cell toxicity).
Minor Comments
- In introduction, facilitate smooth transitions between paragraphs (e.g., link nanotechnology to citrus spoilage more explicitly).
Response: Done, smooth transitions between paragraphs was provided.
- Ensure all data includes error bars and p-values; justify sample sizes (e.g., triplicates).
Response: Done, the required statistical markers were provided, and the sample sizes (e.g., triplicates) were highlighted.
- Table 2 indicates ** and ***, but their purpose was not stated.
Response: Done, the table was revised and corrected.
Reviewer 2 Report
Comments and Suggestions for Authors
This study presents a method for preparing silver nanoparticle-based composites with anti-infective properties. The authors successfully demonstrated particle formation and characterized the composite using various analytical techniques. The antimicrobial performance was also evaluated. Overall, the work offers some advancements in the field of nanocomposites. Publication is recommended, provided the following comments are addressed:
- The rationale for using artichoke as a reducing agent for silver nanoparticle synthesis is not clearly explained in the introduction. Providing more background and context would help better engage and inform the readers.
- The mechanism of silver nanoparticle formation should be more thoroughly discussed in Section 3.2, particularly in relation to the FTIR results. Including a schematic illustration could further aid reader comprehension.
- In Figure 5, it would be helpful to include a comparison of the internal tissue of the oranges to more clearly demonstrate the extent of infection.
- A comparison with other silver nanoparticle-based approaches for anti-infection applications reported in the literature would strengthen the discussion and highlight the novelty of this work.
Author Response
The modified/corrected parts according to Reviewer (2) comments are highlighted with GREEN throughout the manuscript
Responses to Reviewer (2) comments:
- The rationale for using artichoke as a reducing agent for silver nanoparticle synthesis is not clearly explained in the introduction. Providing more background and context would help better engage and inform the readers.
Response: Done, the rationale for using artichoke as a reducing agent for silver nanoparticle synthesis was now clearly explained in the introduction.
- The mechanism of silver nanoparticle formation should be more thoroughly discussed in Section 3.2, particularly in relation to the FTIR results. Including a schematic illustration could further aid reader comprehension.
Response: Done, the mechanism of silver nanoparticle formation was now more thoroughly discussed in Section 3.2, particularly in relation to the FTIR results. A schematic illustration (Graphical abstract) was also provided to further aid reader comprehension
- In Figure 5, it would be helpful to include a comparison of the internal tissue of the oranges to more clearly demonstrate the extent of infection.
Response: Done, the examination of internal tissue of the oranges is difficult to be provided in current stage. The explaination of that was highlighted in the discussion section.
- A comparison with other silver nanoparticle-based approaches for anti-infection applications reported in the literature would strengthen the discussion and highlight the novelty of this work.
Response: Done, results of other former silver nanoparticle-based approaches for anti-infection applications were provided in the discussion.